# A protocol of a pilot randomised trial (Action-RESPOND) to support rural and regional communities with implementing community-based systems thinking obesity prevention initiatives

**Sze Lin Yoong**[1,2]*, **Andrew D. Brown**[1], **Gloria K. W. Leung**[1], **Monique Hillenaar**[1], **Jennifer L. David**[1,3], **Josh Hayward**[1], **Claudia Strugnell**[1,4], **Colin Bell**[1,5], **Vicki Brown**[1,6], **Michelle Jackson**[1], **Steven Allender**[1]

1 Institute for Health Transformation, Global Centre for Preventive Health and Nutrition, School of Health and Social Development, Faculty of Health, Deakin University, Geelong, Australia, **2** National Centre of Implementation Science, College of Health, Medicine and Wellbeing, University of Newcastle, Callaghan, NSW, Australia, **3** School of Public Health, Faculty of Medicine and Health, University of Sydney, Sydney, Australia, **4** Institute for Phaysical Activity and Nutrition, Faculty of Health, Deakin University, Geelong, Australia, **5** School of Medicine, Faculty of Health, Deakin University, Geelong, Australia, **6** Deakin Health Economics, Institute for Health Transformation, Faculty of Health, Deakin University, Geelong, Australia

* s.yoong@deakin.edu.au

## Abstract

### Background

Over a quarter of children aged 2–17 years living in Australia are overweight or obese, with a higher prevalence reported in regional and remote communities. Systems thinking approaches that seek to support communities to generate and implement locally appropriate solutions targeting intertwined environmental, political, sociocultural, and individual determinants of obesity have the potential to ameliorate this. There have however been reported challenges with implementation of such initiatives, which may be strengthened by incorporating implementation science methods.

### Methods

This pilot randomised controlled trial protocol outlines the development and proposed evaluation of a multicomponent implementation strategy (Action-RESPOND). to increase the implementation of community-based systems thinking child obesity prevention initiatives The target of this intervention is ten rural and regional communities (or local government areas as the unit of allocation) within Northeast Victoria who were participants in a whole-of-systems intervention (RESPOND). Action-RESPOND builds on this intervention by assessing the impact of offering additional implementation strategies to five communities relative to usual care. The development of the multicomponent implementation strategy was informed by the Promoting Action on Research Implementation in Health Services (PARIHS) framework and consists of seven implementation strategies primarily delivered via 'facilitation'

**Data Availability Statement:** No datasets were analysed or reported in the current manuscript.

Data sharing is not applicable to this article as no datasets were generated or analysed during the current study.

**Funding:** RESPOND is funded through the National Health and Medical Research Council (NHMRC) (APP115572, CIA SA), VicHealth, Nexus Primary Health, Goulburn Valley Primary Care Partnership and Deakin University. Action-RESPOND is also funded through support funds provided to SY as part of a Heart Foundation Future Leader Fellowship (106654). The opinions, analysis, and conclusions in this paper are those of the authors and should not be attributed to the NHMRC, the Victorian Department of Health or the Victorian Department of Education and Training. SY is supported by a Heart Foundation Future Leader Fellowship (106654). The participating communities were not provided with additional funding to implement the actions.

**Competing interests:** S.A. is a co-inventor of the STICKE software, which is used in this study. All other authors have no conflict of interest to declare. This does not alter our adherence to PLOS ONE policies on sharing data and materials.

methods. Implementation strategies aimed to ensure initiatives implemented are i) evidence-based, ii) address community's specific needs and iii) are suitable for local context. Strategies also aimed to increase the community's capacity to implement, through iv) improving the health promotion team's implementation knowledge and skills, fostering v) leadership, vi) physical resources and vii) community culture to drive implementation. The feasibility, acceptability, potential impact, and cost of the strategy will be assessed at baseline and follow up using surveys administered to key representatives within the community and internal records maintained by the research team.

## Discussion

By leveraging an existing community-based whole-of-systems intervention, Action-RESPOND offers a unique opportunity to collect pilot feasibility and early empirical data on how to apply implementation and systems science approaches to support obesity prevention in rural and regional communities in Victoria.

## Introduction

Childhood obesity is one of the most significant global population health challenges, as it increases children's risk of developing type 2 diabetes, cardiovascular diseases, and liver complications, during their childhood as well as their adulthood [1]. The World Health Organisation estimated that over 39 million children under five and 340 million children aged 5–19 years were overweight or obese in 2016 [2]. In Australia, over 25% of children aged 2–17 years are overweight or obese [3], with rates being disproportionately higher among those living in regional and remote areas [4, 5]. Medical costs related to obesity in children aged 6–13 years are estimated at $ AUD 43 million annually in Australia [6]. It is well established that obesity is caused by a range of environmental and individual determinants that interact in complex systems [7, 8]. For example, broader environmental determinants such as a lack of play spaces or walking paths is likely to reduce an individual's motivation to engage in physical movement and activity, while availability and marketing of 'less healthy' foods could result in poor dietary choices [1].

To date however, much of the empirical research aiming to prevent childhood obesity focuses on interventions that target individuals or single settings. Systematic reviews of interventions targeting single settings, such as schools and childcare centres, show that they can improve dietary outcomes, physical activity and prevent excessive weight gain [9–11], however the effects of these interventions are often small and attenuate when delivered at scale. Reviews of randomised controlled trials (RCTs) of nutrition, physical activity, and obesity prevention interventions report that these interventions retain less than 50% of the effect when scaled up [12–14]. This may be because interventions are often developed and tested in controlled research environments and therefore when scaled up, require adaptation to ensure fit within the real-world context. As such, these interventions do not maintain the same fidelity and dose when scaled up, contributing to the attenuation in effect.

To address such challenges, population-level interventions that move beyond individual interventions and focus on establishing partnerships and systems, which lend themselves to ongoing and sustained lifestyle and environmental changes, are needed to meaningfully prevent obesity [15]. The "Shape Up Somerville" program in Massachusetts (USA) is one such

example of where community-based systems level changes has produced meaningful changes over time [16]. This program involved collaboration between government agencies, schools, local businesses, and community organizations to implement multi-level interventions targeting nutrition, physical activity, and built environment changes between 2003 and 2005. This program produced positive changes on child and parent BMI post intervention with an estimated $197,120 USD net benefits over 10-years. The authors postulated that this positive outcome, which extended beyond improvement to individual children's health, was made possible by the community-wide, systems-based nature of the interventions [16]. Indeed, engaging with these systems forms a key focus of obesity prevention strategies for countries including Australia and the UK [17, 18].

Systems thinking encompasses concepts, principles, and methodologies which allow us to understand, analyse, and address complex problems by considering them as interconnected, dynamic systems rather than isolated components [15]. Some of the key components of systems thinking include feedback loops (i.e. where an action is implemented in the system, and the system responds in a way that either reinforces the initial action taken or pushes back against the initial action), emergent properties (i.e. systems can exhibit emergent properties that arise from interactions and relationships), non-linear causality and dynamic behaviour [19]. A systematic review and meta-analysis found that using system science methods in health service design and delivery significantly improved both patient outcomes (n = 14 studies, OR = 0.52 (95% CI 0.38 to 0.71) $I^2$ = 91%) and service outcomes (n = 18 studies, OR = 0.40 (95% CI 0.31 to 0.52) I2 = 97%) [20]. A narrative review of 65 articles (33 addressing obesity) reported that interventions which incorporated systems science methods resulted in a range of positive health outcomes, although the two good quality no RCTs found mixed effects on health and wellbeing outcomes [21]. Despite the promise of such interventions, there have been well-documented challenges and variability with the implementation of actions arising from these methods [22–24].

In childhood obesity prevention, the Whole of Systems Trial of Prevention Strategies for Childhood Obesity intervention (the WHO STOPS trial) applied a group model building (GMB) process to generate an agreed systems map of childhood obesity causes for a community, and in doing so, identified intervention opportunities through leveraging the dynamic aspects of the system [25]. GMB is a participatory approach grounded in systems thinking and involves a structured process where a diverse group of stakeholders collaboratively create a visual map, to build a shared understanding of the problem and develop community-led, locally tailored solutions [26, 27]. The WHO STOPS trial resulted in reductions in prevalence of overweight/obesity in the first two years of the intervention, however, was not sustained at four-year follow up; due in part to varying intensity of implementation [28]. The Lancet Commission on Obesity noted that inconsistent implementation constrains the effectiveness of community-developed interventions, pointing to the need for a greater application of implementation science to help overcome these barriers [29].

Implementation science [30] provides evidence-based tools, methods, and frameworks to support the design of a more systematic approach to implementation [30] of community-developed, systems thinking-based initiatives. In particular, implementation science draws on empirically-developed frameworks to guide the systematic planning, implementation and evaluation of complex interventions. For example, frameworks such as the Consolidated Framework of Implementation Research [31] and the Theoretical Domains Frameworks [32] have been widely used to systematically identify barriers to implementation and support the selection of evidence-based implementation strategies to improve implementation processes [33]. Intentionally embedding such methods into systems thinking approaches could provide deeper understanding into implementation context, drivers of change and help to

systematically consider the evolving nature of the EBI knowledge users, systems, and organisations within community-based systems approaches [34]. A recent systematic review however, identified just 14 studies that have described the simultaneous use of implementation science and systems science in population health prevention interventions [35]. Of these, few explicitly explored how to apply implementation science to systems thinking approaches, and none explicitly examined how these two sciences could be best utilised together.

Therefore, we sought to undertake a pilot study to describe the potential usefulness of applying implementation science and systems science methods to support delivery of actions generated as part of GMB processes.

Specifically, this pilot RCT aims to:

i. understand the potential impact of implementation strategies on fidelity of implementation;

ii. assess acceptability and feasibility of the implementation strategies;

iii. assess the determinants (barriers and facilitators) related to implementation of identified actions;

iv. document adaptations made to actions in the process of implementation; and

v. quantify the resources required to deliver implementation strategies; and to plan and implement an identified action arising from a participatory community-based systems thinking process (i.e. the GMB process).

## Materials and methods

### Context and setting

This pilot study builds on the Reflexive Evidence and Systems interventions to Prevent Obesity and Non-communicable Disease (RESPOND) study. RESPOND was a cluster randomised controlled trial (cRCT) undertaken with ten rural and regional communities located within the Ovens Murray and Goulburn Valley regions of Northeast Victoria, Australia [36]. Rural and regional classifications were based of that specified by the Victorian Department of Health [37] which is based on the Australian Statistical Geography Standard Remoteness Structure. For this trial, communities were the unit of analysis and defined by local government area (LGA) boundaries, which represent geographical divisions administered by municipal governments within Australian states. RESPOND employed systems thinking approaches in the design, implementation, and evaluation of a community-developed intervention to prevent obesity among children aged 0–12 years.

Five communities were randomised to receive RESPOND and five to a wait-list control. The methods for this study have been described in detail elsewhere [36] and the intervention included: i) establishment of strong governance with key partners (including Beechworth Health Service, Central Hume Primary Care Partnership, Gateway Health, Goulburn Valley Primary Care Partnership, Greater Shepparton City Council, Lower Hume Primary Care Partnership, Numurkah District Health, Upper Hume Primary Care Partnership, VicHealth, the Victorian Department of Education and Training, the Victorian Department of Health and Human Services and Yarrawonga Health), and clearly define roles and responsibilities; ii) routine childhood obesity and risk factor monitoring to support outcome assessment; and iii) capacity building to the communities in systems science methods, notably facilitating participatory systems mapping methods (i.e. the GMB processes) and iv) implementation of actions. All ten communities received the RESPOND program between 2020–2023. The delivery of the RESPOND program coincided with significant disruptions due to COVID-19 and bush fires

in the region, significantly impacting on organisational priorities, loss of momentum of implementation, and redeployment of human resources [24].

## Trial design

This pilot RCT (called Action-RESPOND) seeks to build on some of the implementation challenges faced in RESPOND by systematically developing an enhanced implementation support package and assessing the feasibility and potential impact of this with the ten communities already participating in RESPOND. Five communities will receive a multicomponent implementation strategy (intervention) and five will not receive any additional implementation support (control). The trial will assess data at baseline and approximately 9-months follow up (see Fig 1). The reporting of this protocol was consistent with the CONSORT-pilot RCT guidance [38] and the Standards for Reporting Implementation Studies (STARI) statement [39]. Data collection for this trial began in June 2023.

## Ethics

This study has received ethical approval from Deakin University Human Research Ethics Committee (HEAG-H 12_2019). Written informed consent will be obtained from all participants prior to each data collection activity (via an online consent form prior to the survey or during attendance at the GMB workshops by a member of the research team).

## Eligibility criteria

The overall inclusion criteria for this study is at the 'community' level (i.e. LGAs in Victoria). Communities that are included within the geographical area considered as rural/regional and have agreed to participate in the larger RESPOND Trial will be invited to participate. Ten LGAs in Victoria are included.

The backbone team from each LGA form the main participants in the current study. Recruitment of the LGAs and backbone team members were undertaken as part of the larger RESPOND Trial. For members to be part of the current study, they need to have some formal role in implementation (and could report on extent of implementation) and are located in the target LGAs.

## Randomisation

Randomisation occurred at the level of whole communities (LGAs) as part of the larger RESPOND trial and was conducted by a statistician with no knowledge of the communities. Ten LGAs were ranked in order of population size and divided into five pairs. A computer-generated random list was created by a study statistician who was not involved in the enrolment procedures. One community from each pair received either Action-RESPOND or usual care using a random list generated by the study statistician.

## Blinding

Action-RESPOND is an open trial, and no blinding was possible due to the nature of the intervention. Therefore, those receiving the intervention were not blinded to group allocation; however, the statistician conducting the data analysis will remain blinded.

## Intervention

**The systems thinking child obesity prevention initiative (RESPOND trial).** As part of RESPOND, all communities received eight support sessions to build their capacity to deliver a

| STUDY PERIOD | | | | | | | |
| --- | --- | --- | --- | --- | --- | --- | --- |
| | Allocation | Enrolment | Post-allocation | | | | Close-out |
| Timepoint | *RESPOND* | *Baseline* | *GMB3[1] debrief* | *Edu Workshop* | *Check-in #1* | *Check-in #2* | *Follow-up* |
| **Enrolment:** | | | | | | | |
| Completion of RESPOND GMB3[1] | | X | | | | | |
| Informed consent | X | | | | | | |
| Allocation | X | | | | | | |
| **Action-RESPOND Interventions:** | | | | | | | |
| Provision of evidence-briefs | | | X | X | | | |
| Support prioritisation of evidence-based and locally appropriate innovations | | | | X | | | |
| Establish processes to feedback to and engage community | | | | | X | X | |
| Selection of Implementation Champion/s | | | | X | | | |
| Goal setting | | | | X | | | |
| Identify determinants of successful implementation of action | | | | X | | | |
| Action planning | | | | X | X | | |
| Tailor facilitation approach to individual communities | | | | X | X | X | |
| Inform and engage leadership | | | | X | X | X | |

| STUDY PERIOD | | | | | | | |
| --- | --- | --- | --- | --- | --- | --- | --- |
| | Allocation | Enrolment | Post-allocation | | | | Close-out |
| Timepoint | *RESPOND* | *Baseline* | *GMB3[1] debrief* | *Edu Workshop* | *Check-in #1* | *Check-in #2* | *Follow-up* |
| **Assessments:** | | | | | | | |
| Fidelity & adoption of action implementation (via survey) | | X[2] | | | | | X[2] |
| Acceptability of implementation strategies (via survey) | | | | | | | X |
| Appropriateness and usefulness of the implementation strategies (via survey) | | | | | | | X |
| Cost (via internal project records) | | | | | | | X |
| Changes in implementation determinants and adaptations to actions (via STICKE software) | | | | | X | X | X |

[1]GMB3, the third group model building workshop in the RESPOND trial.
[2]Completed by both intervention and control communities of Action-RESPOND.
STICKE, Systems Thinking in Community Knowledge Exchange software.

**Fig 1. Schedule of enrolment, intervention and assessments for the Action-RESPOND pilot RCT as outlined in the SPIRIT recommendations [39].**

series of three GMB workshops locally with key stakeholders in their communities. GMB is a qualitative, facilitated process, where a range of stakeholders create a visual map to build a shared understanding of obesity. The visual diagrams built as part of this process are called

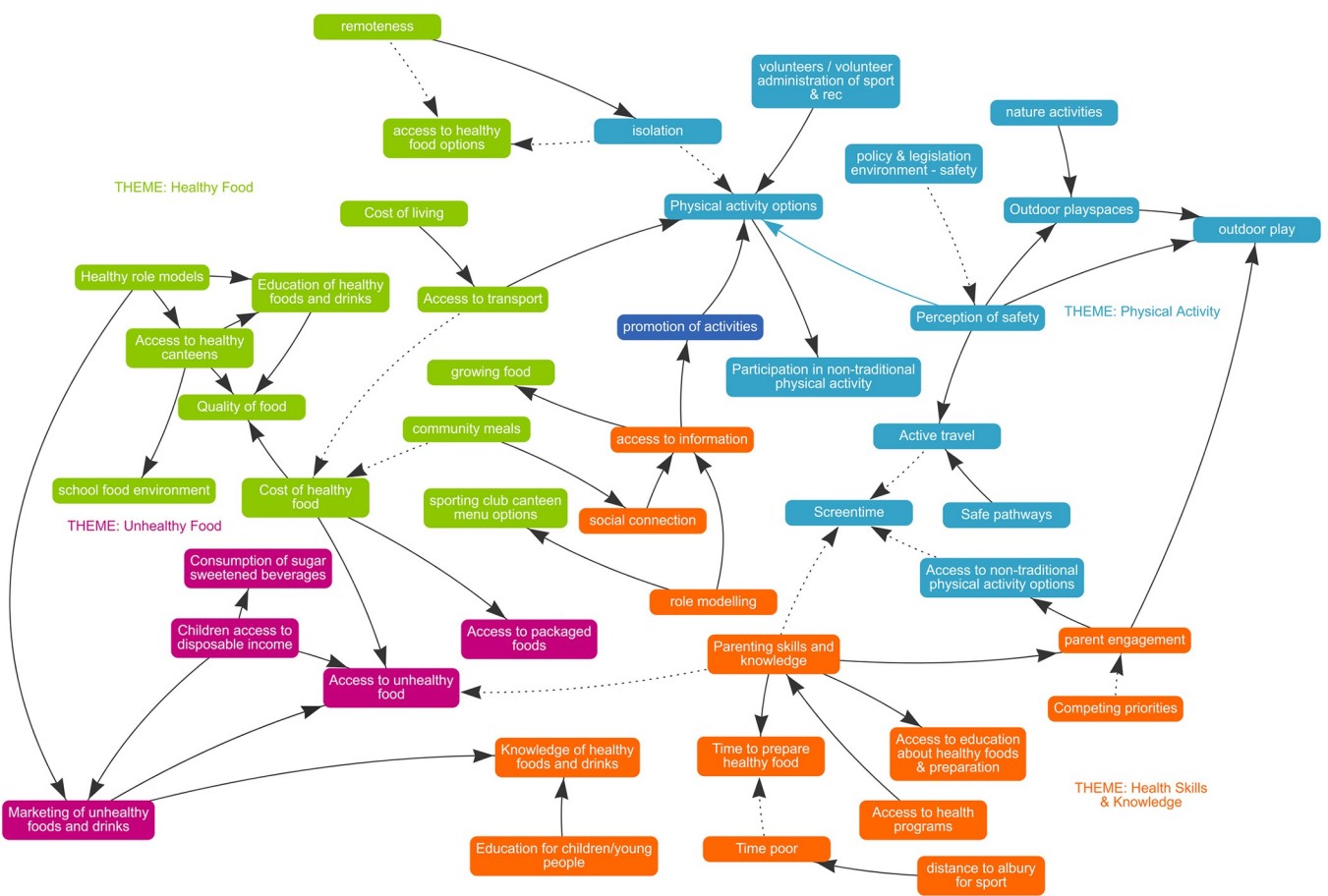

**Fig 2. Example causal loop diagram (CLD) from RESPOND built in partnership by local organisations and community stakeholders from Towong Shire, Victoria, Australia on the factors driving community health.**

causal loop diagrams (CLDs) and they were developed on a software created by the research team (STICKE; Systems Thinking in Community Knowledge Exchange) [40] (see Fig 2 for an example). CLDs represent the variables participants perceive to be contributing to a problem and the causal connections between them. The CLDs are then used to help identify action ideas that are contextually appropriate to their community [22].

The GMBs are facilitated by the 'backbone team' within each community, who members are defined as those who have the authority and responsibility to make changes that affect the determinants of health within the community. The 'backbone team' consists of a combination of community development workers from local councils, health promotion practitioners from relevant health services and relevant non-government organisations in the region. By the end of the third GMB workshop (i.e., GMB3), multiple actions to address the identified variables in the CLD are generated by participants and these form the target of the implementation strategies below. The specific actions varied from community to community, however they broadly targeted community efforts to improve physical activity and nutrition in young children. Examples of some action ideas included community gardens, school-based cooking classes, efforts to improve the built environments (i.e. water fountains, paths), and increasing community participation and exposure to different sporting events and sporting infrastructure.

## The multicomponent implementation strategy (Action-RESPOND)

*Theoretical approach*. We applied the PARIHS (Promoting Action on Research Implementation in Health Services) framework [41] to support planning and development of the multicomponent implementation strategy (Action-RESPOND). This framework was selected as it focuses on capacity building and iterative tailoring of support, which is consistent with community-led approaches. PARIHS suggests that successful implementation occurs where there is a strong alignment between evidence, context and facilitation [41]. Specifically, it indicates that development of actions should consider 'evidence' that is contextually relevant and appropriate to the community, and implementation support is primarily delivered via a skilled 'facilitator/coach', who is familiar with the unique challenges and needs of the community (i.e. 'local context') and adapts support accordingly [41].

Using the PARIHS framework, we identified key determinants and opportunities to support implementation of actions at the community level. This was done by undertaking a targeted search of the literature and drawing on previous interviews with communities [35, 42, 43]. From this, we mapped the findings to the domains of the PARIHS framework (see Table 1) and identified the following key determinants of implementation: i) Limited knowledge of the research evidence underpinning selected actions; ii) ensuring implemented action addresses the community's preferences and needs; iii) Ensuring the evidence-based action is compatible with the community's context and needs; iv) Lack of knowledge and skills on how to implement actions developed using systems thinking approaches, i.e. perceived feasibility and difficulty of implementation, and steps required to implement; v) Cultivating leadership that is supportive of implementation of action; vi) Having human and other resources to drive implementation in local community; and vii) Establishing community culture that is receptive to and supportive of implementation of action.

*Delivery modality*. Action-RESPOND consist of seven implementation strategies that will be delivered over approximately 9-months via one online meeting (GMB3 debrief), one face-to-face group education workshop (Edu Workshop), two formal facilitation contacts delivered by an implementation coach (Check-in sessions) and local tailored support as requested. Additionally, we will incorporate system science concepts, including qualitative systems mapping, use of reference modes (which capture the dynamic nature of key factors within a system) and feedback loops, during the delivery of the multicomponent implementation strategy (see Table 1). The research team delivering the intervention consists of systems science experts (AB, JH) and implementation scientists (SY, GKWL) and an Implementation Coach (MH). The Implementation Coach (MH) has 10 years of health promotion experience working within the targeted communities and is situated within one community of the Ovens-Murray region.

The Edu Workshop will be delivered face-to-face to the 'backbone team' who consist of community development workers from local councils, health promotion practitioners from relevant health services and relevant non-government organisations in the region. This session will be co-facilitated by systems science (AB) and implementation science (SY) experts together with the Implementation Coach (MH).

The Implementation Coach (MH) will be primarily responsible for delivering the two facilitation check-in sessions. The sessions will either be delivered over videoconference call or face-to-face (depending on preference of community) with identified Implementation Champion(s) within each community. The Champion/s are likely to consist of one or multiple individuals (in the instance of shared role or part-time staff) who are health promotion officers within the local community health teams or local council.

*Description of implementation strategies*. Consistent with the PARIHS framework, we will primarily use an implementation facilitation approach to deliver specific strategies.

**Table 1. Determinants to implementation identified in the evidence and context domains of the Promoting Action on Research Implementation in Health Sciences (PARIHS) framework, with corresponding implementation strategies and details of delivery.**

| Determinants (as identified using the PARIHS framework) | Implementation Strategy (Action)[1] | Implementation Strategy delivery [a]: <br> • *Actor (who is delivering)* [b] <br> • *Context (delivery method)* <br> • *Target (who is receiving it)* [b] <br> • *Time (duration & frequency of delivery)* |
|---|---|---|
| **Evidence** | | |
| Limited knowledge of the research evidence underpinning selected actions. | Increasing awareness and knowledge of research evidence supporting the effectiveness of actions generated during GMBs: <br> i) Evidence brief presentation on actions generated in GMBs, which the community is most interested in. <br> ii) Evidence summary on community-based childhood obesity prevention interventions | Actor: research team |
| | | Context: <br> i) Presentation during GMB3 debrief (online meeting) <br> ii) Recorded presentation |
| | | Target: backbone team |
| | | Time: <br> i) After GMB3; approx. 15-minute presentation during 2-hour meeting <br> ii) Between GMB3 and Edu Workshop; approx. 20-minute presentation |
| Ensuring implemented action addresses the community's preferences and needs. | i) Multiple sense-checks during selection of action and goal setting, to ensure it aligns with community's preferences (as expressed by community members during GMBs) <br> ii) Embedding processes in action plan, to involve community members in implementation of action and to feedback progress of action to wider community | Actor: research team |
| | | Context: <br> i) Edu Workshop—action prioritisation and SMART goal planning activity <br> ii) Edu Workshop—Action planning activity |
| | | Target: backbone team |
| | | Time: <br> i) Edu Workshop (face-to-face, 6-hour) <br> ii) Edu Workshop (face-to-face, 6-hour) and prompt during check-in sessions |
| Ensuring the evidence-based action is compatible with the community's context and needs. | Building capacity to prioritise and select action with consideration of research evidence and local context information, using the Hexagon Tool [c] | Actor: research team |
| | | Context: Edu Workshop—action prioritisation activity |
| | | Target: backbone team |
| | | Time: Edu Workshop (face-to-face, 6-hour) |
| Lack of knowledge and skills on how to implement actions developed using systems thinking approaches, i.e. perceived feasibility and difficulty of implementation, and steps required to implement. | i) Building capacity to identify locally relevant enablers and barriers to implement selected priority action <br> ii) Increase knowledge on implementation strategies and support action planning <br> iii) Support problem solving (tailored to each individual community) during implementation process | Actor: <br> i) Research team <br> ii) Research team <br> iii) Implementation Coach |
| | | Context: <br> i) Edu Workshop–change over time and implementation CLD building activity <br> ii) Edu Workshop–provision of training and resources (implementation tools and action plan template) and demonstrate application <br> iii) During check-in sessions |
| | | Target: <br> i) Backbone team <br> ii) Backbone team <br> iii) Implementation Champion/s |
| | | Time: <br> i) Edu Workshop (face-to-face, 6-hour) <br> ii) Edu Workshop (face-to-face, 6-hour) <br> iii) Check-in sessions (online meeting, 1.5 hours, approx. 2-months apart) |
| **Context** | | |

(*Continued*)

**Table 1.** (Continued)

| Determinants (as identified using the PARIHS framework) | Implementation Strategy (Action)[1] | Implementation Strategy delivery [a]: • *Actor (who is delivering)* [b] • *Context (delivery method)* • *Target (who is receiving it)* [b] • *Time (duration & frequency of delivery)* |
|---|---|---|
| Cultivating leadership that is supportive of implementation of action. | Inform and engage key decision-makers, including health service managers and local council members | Actor: i) Implementation Coach ii) Research team lead |
| | | Context: i) Invite key decision-makers to attend Edu Workshop (via email) ii) Quarterly presentation to RESPOND leadership/ stakeholder group, providing update on Action-RESPOND |
| | | Target: i) Key decision-makers in the community ii) Key decision-makers in the community |
| | | Time: i) Prior to Edu Workshop ii) RESPOND Partnership Group Meeting (online, occurs quarterly) |
| Having human and other resources to drive implementation in local community. | Identify and formalise role of Implementation Champion in each community | Actor: Implementation Coach |
| | | Context: Position description circulated via email and formalise at Edu Workshop |
| | | Target: Backbone team |
| | | Time: Edu Workshop |
| Establishing community culture that is receptive to and supportive of implementation of action. | i) Involve community members in the GMB process ii) Ensure continued feedback to community regarding progress of action implementation | Actor: i) Research team ii) Implementation Coach and Implementation Champion |
| | | Context: i) GMB workshops ii) Establishing these processes during Check-in sessions |
| | | Target: i) Community members and stakeholders ii) Community members and stakeholders |
| | | Time: i) GMB workshops ii) Check-in sessions (online meeting, 1.5 hours, approx. 2-months apart) |

[a] Based on Presseau et al 2019.

[b] Actors and Targets: a) Backbone team–comprised of community development workers from local council, health promotion practitioners from local health services/ non-government organisations in each community; b) Implementation Champion/s–identified as key representative from the backbone team; c) Research team– comprised of systems science experts (AB, JH), implementation scientists (SY–team lead; GKWL) and Implementation Coach (MH); d) Implementation Coach (MH)– situated in and has extensive experience working in health promotion within targeted communities.

[c] Based on Metz et al 2019.

Abbreviations: GMB–group model building; GMB3 –the 3rd group model building workshop of the RESPOND trial; CLD–causal loop diagram.

Implementation facilitation is a multi-faceted process to enable communities to develop processes, relationships and structures to increase implementation and address key identified local gaps [44]. The selection of implementation strategies was overseen by an advisory group consisting of implementation scientists, system scientists, and health promotion and community development experts. The strategies were intentionally selected to address the

determinants identified in Table 1, based on evidence of effectiveness from a systematic review of facilitation strategies [45] and described using the Action, Actor, Context, Target, and Time framework [46]. The implementation strategies were piloted with one community and amendments made to refine content. The refined implementation strategies, which will be delivered, are described below.

## 1. Generation and provision of evidence-briefs and local data

Following GMB3, a presentation of the evidence for the actions that have emerged will be provided by the research team. This will be accompanied by local data on potential reach of such actions and the extent of local implementation where available. This will be briefly presented at an online meeting and a discussion around the actions (GMB3 debrief), facilitated by a member of the research team. Additionally, prior to the Edu Workshop, a recorded Power-Point presentation summarising reviews on effectiveness of community-based childhood obesity prevention interventions are circulated to the backbone team of each community.

## 2. Support prioritisation of evidence-based and locally appropriate innovations

As many actions are generated via the GMBs, communities will be subsequently supported to prioritise actions for implementation. At the Edu Workshop, the Hexagon Tool [47] will be used to support the prioritisation of actions identified. Prioritisation will be achieved by a discussion around the following elements regarding their key actions: capacity to implement, the evidence surrounding the intervention, fit with current policy environment and existing initiatives, how well it addresses local needs, usability and existing support systems. This prioritised action will form the behavioural target of the subsequent implementation strategies. Although the backbone team will be asked to prioritise and focus on a single action, it is acknowledged that other actions led by community members or groups may also be implemented at the same time, recognising the broader community engagement with this process.

## 3. Establish processes to feedback action to community and continue to engage community members

Building on the inherently participatory processes of the GMBs and existing community engagement, the Implementation Coach will continue to work with Implementation Champion/s (see #4) to establish mechanisms to continue to engage with community members. This may include timely reporting on the key actions following GMB processes, communicating an update on implementation activities, and creating feedback mechanisms as appropriate, considering reach and accessibility (i.e., follow up emails, surveys, social media updates, newsletters, reports). Each community will be supported to develop a communication plan.

## 4. Selection of implementation champion/s

The use of champions can enhance implementation through various mechanisms including trust-based relationships, role modelling and advocacy [48]. An Implementation Champion (or multiple) will be identified by each community's backbone team prior to the Edu Workshop. A position description for the Champion will be provided outlining the expectations for the role. This includes supporting the development and implementation of an action plan (see 5c), mapping the implementation processes on STICKE and monitoring implementation progress.

## 5. Provide training that is dynamic and applied

The Edu Workshop will be undertaken with each community's backbone team. The training will specifically target barriers related to knowledge and skills identified by health promotion teams within the communities and focus on increasing capacity to support implementation of the prioritised action. The session is designed to be interactive; incorporating group discussions, real world examples and Group Model Building activities around implementation of the priority action. It will include the following strategies:

5a. Goal setting

The research team will support the participants with generating a SMART (Specific, Measurable, Achievable, Relevant and Time-bound) goal related to the prioritised action, that is considered timely, practical and feasible to achieve in the next 6–9 months. This will be transferred across to their action plan (see 5c).

5b. Qualitative systems methods to visualise the processes of change, support participants to identify determinants of successful implementation of action

The systems science expert (AB) will undertake a change over time activity to support participants to visualise how changes in the prioritised action would have looked like in the past, in the future, and if active steps were /were not taken to implement the action. The resulting "reference mode" is a visual method used in system science, describing how a key outcome of interest behaves or operates under different conditions, or in response to various inputs or changes in the context of a system's structure (I.e., policies, culture, and infrastructure) [19]. This will be used to support the identification of determinants of successful implementation of action (i.e., what needs to be done to avoid/achieve such trends). This is supplemented further by applying relevant implementation science theories. The specific theory will be determined by the implementation science expert (SY) pending action selection and community context, however could include the Theoretical Domains Framework [32], the Consolidated Framework of Implementation Research [31] or the EPIS framework [49]. Once the barriers have been identified and prioritised, the group will be supported to generate a systems map using the STICKE software, creating a CLD outlining determinants to implementation. This will reflect the adaptive nature of implementing action, drawing feedback loops to describe how a system responds to changes in conditions resulting from implementing action/s.

5c. Action planning

Lastly, the training will support participants with identifying specific strategies to address key determinants and this will be used to populate an action plan. The action plan will outline the specific activity, who will be responsible for implementation, the timeframe for implementation, and the resources required. Both the CLD for implementation and the action plan developed during this education session will be visually presented on STICKE and used in the Check-in sessions to support tracking of activities in the action plan.

## 6. Tailor facilitation approach to individual communities

In the Check-in sessions with the Implementation Champion/s, the Implementation Coach will support the ongoing assessment of implementation context, tailoring of the intervention to suit local context, ongoing monitoring of progress and outcomes, and identify ways to continue community engagement. The Check-in session will begin with a review of the implementation CLD generated in the Edu Workshop, to track progress made with each action plan activity, and record any adaptations made to the action plan. Throughout the session, the Implementation Coach will provide positive reinforcement, facilitate reflection and problem solving. The discussion will be an opportunity to update the action plan, ensuring that it contains required activities to achieve the SMART goal and set new activities, goals or actions if

required. The support will be tailored according to the preferences, needs and/or barriers of each community. The Implementation Coach is available to be contacted for additional local tailored support throughout the intervention period. All support provided including each contact, attendees, duration and nature of support will be captured using a Microsoft Excel spreadsheet.

## 7. Inform and engage leadership

Health promotion, allied health and relevant local council managers will be invited to the GMBs and face-to-face Edu Workshop. This is to generate leadership support for the process, interventions and decisions around action planning made by the backbone team. Further, quarterly updates will be provided by the research team lead (SY) to the leadership group around the progress of the activities related to the research, with an attempt to support buy in and promotion of the Action-RESPOND program. The leadership group consists of the research team along with key RESPOND partner organisations, including the Department of Health, the Victorian Department of Education, management from local community health services, VicHealth and regional Local Council representatives.

*Control group*. Communities in the control group will receive usual care as part of RESPOND, including regular updates, networking opportunities, and implementation support as requested via e-mail and phone.

## Outcomes

The study will assess the following outcomes to address the main aims. This will be collected at baseline for both groups (before receiving implementation support) and at follow-up (approximately 9-months post intervention). Data collection for all outcomes listed below will be conducted via an online survey (Qualtrics, Utah, United States), unless otherwise stated.

## Fidelity and adoption of action implementation

The backbone teams of both the intervention and control communities will select one representative to report on the fidelity of action implementation at baseline and at follow-up. Although many actions may be identified as part of the systems process, they will be asked to describe one selected action for implementation. Fidelity is assessed via three items with responses on a Likert scale, reporting on i) the extensiveness (1 = not at all to 7 = extensively), and ii) frequency (1 = not at all to 7 = all the time) of action implementation and iii) how engaged (1 = not at all to 7 = all the time) organisation and individuals are with the implementation process. Additionally, the survey will assess adoption of the selected action, which will be measured using three items assessing 'intention to implement' the action (on a Likert scale, with 1 = Strongly disagree to 7 = Strongly agree). These items were adapted from previous validated items used by the research team [50].

## Acceptability of implementation strategies

All backbone team members of intervention communities (between 3–8 in each community) will be asked to report on the acceptability of the intervention at follow-up. Acceptability will be defined as the perception amongst the backbone team that the intervention and implementation strategies are agreeable, palatable or satisfactory [51]. This will be assessed using seven items on a five-point Likert scale reporting on the following elements of Action-RESPOND:

- Affective attitude (1 = strongly dislike to 5 = strongly like),

- Burden (1 = no effort at all to 5 = huge effort),

- Perceived effectiveness (1 = strongly disagree to 5 = strongly agree),

- Intervention coherence (1 = strongly disagree to 5 = strongly agree),

- Self-efficacy (1 = not at all confident to 5 = very confident),

- Opportunity costs (1 = did not have to give up activities to 5 = gave up many activities), and

- Overall acceptability (1 = completely unacceptable to 5 = completely acceptable).

- These items are underpinned by the Theoretical Framework of Acceptability [51], a validated framework used to assess the acceptability of healthcare interventions from the perspectives of intervention deliverers and recipients.

### Appropriateness and usefulness of the implementation strategies

At follow-up, the appropriateness of the implementation strategies will be assessed with intervention communities via four items on a Likert-scale (1 = strongly disagree to 5 = strongly agree) and one open-ended item. The backbone team members will also be asked to report on how useful they thought each of the individual implementation strategies were (Likert scale, with 1 = not helpful at all to 5 = extremely helpful) and to qualitatively report on why they rated the components this way.

### Cost

The total cost of delivering the Action-RESPOND program will be estimated, including labour (i.e., research team time, facilitator preparation, administration, and delivery of the program), travel and equipment. A costing tool will be programmed in Microsoft Excel and used to track program delivery costs according to opportunity cost principles. The cost for communities to receive the program will be estimated using administrative records (e.g., attendance records at Edu-workshop and Check-in sessions). The time, travel and equipment cost of supporting the implementation of the prioritised action will be estimated using survey data. Published wage rates will be used to estimate labour costs, and travel and equipment costs will be estimated using market rates.

### Changes in implementation determinants and adaptations to actions

This will be captured in the intervention communities only, via the STICKE software, and include structured data collection (including specific prompts and questions) as part of the Check-in sessions. This can be exported into a Microsoft Excel spreadsheet and will be collated by the research team following each contact with the Implementation Champion of intervention communities. This will capture how existing barriers change, any new barriers and the relationship to the existing barriers, action progress and strategies implemented.

### Potential co-intervention and contamination

At follow-up, we will assess if communities received any other support or new funds to implement their actions (outside of the research team) and to describe this support. Additionally, we will assess the extent of contamination in participants of the control communities, by asking whether they have received any component of the Action-RESPOND program and any other support to deliver the proposed initiatives.

## Co-benefit and adverse outcomes

At follow-up, we will assess if any additional actions were implemented resulting from receiving Action-RESPOND and for participants to describe the specific actions.

We will ask of any other unintended outcomes (both positive and negative) observed by the communities as a part of participating in this trial.

## Sample size

As this is a pilot study, a formal sample-size calculation was not undertaken in line with best practice guidance [38]. However, 10 communities (with between 3–8 representatives) are included, as this number was considered feasible in the timeframe and allocated resources and is sufficient to provide an adequate indication of the feasibility and acceptability of study methods and the sustainment strategy.

## Data management

All data will be collected online and directly entered into the survey platform, and as such no data entry errors are expected. The majority of survey items are close-ended in response and require a response to minimise missing data. The research team will send reminders and follow up on any non-responses to minimise missing data. Given the pilot nature and the small number of participants, a manual check will be undertaken to identify any clear incongruence in responses. If such responses are identified, this will be discussed with the data monitoring committee and relevant actions will be made depending on the type of data and extent of misreporting. This includes clarifying with participants, and/or using previously reported relevant data to inform data imputations. All decisions will be documented and reported in the analysis process. The data collected will be stored on a secured server at Deakin University in accordance with the requirements of all ethics committee. The data monitoring committee comprises of the chief investigators on the grant and the Action-RESPOND team. Personal information of participants (name and contact details) will be stored in a password-protected electronic Excel Spreadsheet. Only research staff approved on the ethics application will have access to this file or any raw data of the study. All results will be reported at the unit of the LGA. The Action-RESPOND team will monitor the progress of the trial, oversee the data collection progress, and provide updates to the data monitoring committee. Reports will be submitted annually to the ethics committee and funding body.

## Data analysis

All statistical analyses will be undertaken using relevant statistical software. Descriptive statistics (mean and standard deviation (SD) or median and range when non-normal) will be generated for all assessed outcomes. For acceptability, feasibility, appropriates and usefulness, individual scores from each item will be summed (reverse scored where necessary) and combined to provide a score for each community. A score of $\leq 2$ will be considered a negative response, indicating the Action-RESPOND program is considered unacceptable, while $\geq 4$ will be indicative of an acceptable program. For feasibility, $\geq 4$ (higher scores) will indicate greater feasibility, while $\leq 2$ (lower scores) will indicate lower feasibility, consistent the recommendations of Weiner et al (2017) in the Feasibility of Intervention Measure [52]. Answers to the open-ended questions will be analysed qualitatively.

For implementation outcomes (fidelity and adoption of the selected action for implementation), item responses will be summed to provide an overall fidelity and an overall adoption score for both the intervention and control group. This will be compared between groups

using a linear regression, controlling for baseline scores. Changes in implementation determinants and adaptations to actions will be described narratively for each community.

## Progression criteria

The advisory group for this research will assess suitability for progression to a fully powered trial in similar communities in Victoria, using data collected related to fidelity, adoption, and acceptability of the implementation strategies via this study as well as systematic reviews of systems-based obesity prevention interventions on the potential benefits on child outcomes. Together this data will allow for assessment on the potential for a larger trial focused on implementation specifically. These decisions will be made via majority, by core members of the research team, including a representative from participating communities. Specifically, the team must rate the program as sufficiently acceptable and feasible for it to be likely adopted by over half of the communities in which it was offered. Alternatively, the team may decide that this could be reasonably expected with adaptations to the program, based on steps previously employed. Further, the data collected will be used to identify opportunities to strengthen the intervention and refine trial methods prior to a fully powered implementation trial.

## Trial discontinuation or modification

It is not anticipated that any events would occur that would warrant discontinuing the trial. Baseline data collection for Action-RESPOND has commenced, and the communities have been recruited to the RESPOND trial and continue to be engaged with the program. Any unforeseen adverse events will be reported to Deakin HREC (primary approval committee) and appropriate action taken to address the event. The trial registration record will be updated with any protocol modifications and any deviations from the original protocol will be reported when publishing trial outcomes.

## Dissemination plan

The lead author together with the advisory group will develop and oversee the project dissemination plan including all publications and reports to stakeholders. Authorship will conform to the International Committee of Medical Journal Editors (ICMJE) guideline. The results of the Action-RESPOND trial will be published in the academic literature, in conference presentations and in publications associated with postgraduate research projects. Relevant presentations and any other stakeholder reports will be developed upon request from the research partners on this trial, including via existing network meetings consisting of all LGAs and partners of the research.

## Discussion

This protocol outlines methods to formally combine systems and implementation science approaches for obesity prevention, and for the first time seeks to pilot such approaches in a RCT design. Despite calls to better understand how systems science and implementation science can be collectively used to improve the impact of public health programs [29, 35], there is lack of empirical studies addressing this, hindering our ability to fully harness the potential of these disciplines in addressing complex real-world challenges. By building on an existing intervention and responding to an identified need expressed by communities, Action-RESPOND seeks to address these gaps and provide data about the feasibility, acceptability, and potential outcomes of such an approach on maximising the impact of community-generated systems initiatives to prevent childhood obesity.

## Other information

### Registration

This study was retrospectively registered with ANZCTR (ACTRN12623000719639) as one community had commenced baseline assessment of Action-RESPOND prior to approval.

## Supporting information

**S1 Checklist. SPIRIT 2013 checklist: Recommended items to address in a clinical trial protocol and related documents\*.**
(DOC)

**S1 File.**
(DOCX)

## Author Contributions

**Conceptualization:** Sze Lin Yoong, Andrew D. Brown, Gloria K. W. Leung, Steven Allender.

**Data curation:** Andrew D. Brown.

**Funding acquisition:** Sze Lin Yoong, Claudia Strugnell, Colin Bell, Steven Allender.

**Investigation:** Sze Lin Yoong, Andrew D. Brown, Gloria K. W. Leung, Monique Hillenaar, Jennifer L. David, Josh Hayward, Colin Bell, Michelle Jackson.

**Methodology:** Sze Lin Yoong, Andrew D. Brown, Gloria K. W. Leung, Monique Hillenaar, Jennifer L. David, Josh Hayward, Claudia Strugnell, Vicki Brown.

**Project administration:** Gloria K. W. Leung, Monique Hillenaar, Jennifer L. David.

**Resources:** Sze Lin Yoong.

**Supervision:** Sze Lin Yoong.

**Writing – original draft:** Sze Lin Yoong, Andrew D. Brown, Gloria K. W. Leung.

**Writing – review & editing:** Monique Hillenaar, Jennifer L. David, Josh Hayward, Claudia Strugnell, Colin Bell, Vicki Brown, Michelle Jackson, Steven Allender.

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
