## [Decision Letter · Decision Letter 0]

30 Jan 2024

PONE-D-23-35824A protocol of a pilot randomised controlled trial (Action-RESPOND) to support rural and regional communities in Victoria, Australia, with implementing community-based systems thinking child obesity prevention initiatives.PLOS ONE

Dear Dr. Yoong,

Thank you for submitting your manuscript to PLOS ONE. After careful consideration, we feel that it has merit but does not fully meet PLOS ONE’s publication criteria as it currently stands. Therefore, we invite you to submit a revised version of the manuscript that addresses the points raised during the review process. The paper has been assessed by two peer reviewers. While the statistical reviewer was happy with the paper's statistics, the clinical reviewer has highlighted a number of issues needing clarification.

We look forward to receiving your revised manuscript.

Kind regards,

Sascha Köpke

Academic Editor

PLOS ONE

Journal Requirements:

"RESPOND is funded through the National Health and Medical Research Council (NHMRC) (APP115572, CIA SA), VicHealth, Nexus Primary Health, Goulburn Valley Primary Care Partnership and Deakin University. Action-RESPOND is also funded through support funds provided to SY as part of a Heart Foundation Future Leader Fellowship (106654). The opinions, analysis, and conclusions in this paper are those of the authors and should not be attributed to the NHMRC, the Victorian Department of Health or the Victorian Department of Education and Training. SY is supported by a Heart Foundation Future Leader Fellowship (106654). The participating communities were not provided with additional funding to implement the actions."

"S.A. is a co-inventor of the STICKE software, which is used in this study. All other authors have no conflict of interest to declare. "

5.PLOS requires an ORCID iD for the corresponding author in Editorial Manager on papers submitted after December 6th, 2016. Please ensure that you have an ORCID iD and that it is validated in Editorial Manager. To do this, go to ‘Update my Information’ (in the upper left-hand corner of the main menu), and click on the Fetch/Validate link next to the ORCID field. This will take you to the ORCID site and allow you to create a new iD or authenticate a pre-existing iD in Editorial Manager. Please see the following video for instructions on linking an ORCID iD to your Editorial Manager account: https://www.youtube.com/watch?v=_xcclfuvtxQ

6. We note that the original protocol that you have uploaded as a Supporting Information file contains an institutional logo. As this logo is likely copyrighted, we ask that you please remove it from this file and upload an updated version upon resubmission.

Reviewers' comments:

Reviewer's Responses to Questions

**Comments to the Author**

1. Does the manuscript provide a valid rationale for the proposed study, with clearly identified and justified research questions?

Reviewer #1: Yes

Reviewer #2: Yes

2. Is the protocol technically sound and planned in a manner that will lead to a meaningful outcome and allow testing the stated hypotheses?

Reviewer #1: Yes

Reviewer #2: Yes

3. Is the methodology feasible and described in sufficient detail to allow the work to be replicable?

Reviewer #1: Yes

Reviewer #2: Yes

4. Have the authors described where all data underlying the findings will be made available when the study is complete?

Reviewer #1: Yes

Reviewer #2: No

5. Is the manuscript presented in an intelligible fashion and written in standard English?

Reviewer #1: Yes

Reviewer #2: Yes

6. Review Comments to the Author

You may also provide optional suggestions and comments to authors that they might find helpful in planning their study.

Reviewer #1: Thank you for the opportunity to review this paper.

My review mainly concerns only the statistical aspects of the study.

The paper is well write and the simple statistical analysis plan well described

Reviewer #2: Thank you for the opportunity to review this paper describing the protocol of a randomized controlled trial to address child obesity in Victoria Australia. I commend the authors for developing a well-written manuscript and for their important efforts to address childhood obesity. While the manuscript is well-written and addresses a very important topic, there are several areas where the manuscript can be strengthened, which may also have implications for the Action-RESPOND trial. I offer the following suggestions below to strengthen the manuscript and its contribution to the literature. I hope the authors find my feedback to be helpful and constructive.

TITLE

1. As written, the title is rather lengthy. Can the title be shortened a bit?

ABSTRACT

2. In the methods section of the abstract, should the first mention of “(Action-RESPOND”) appear after the words “multicomponent implementation strategy…”? It’s a bit confusing to figure out of the intervention is the implementation strategy or community-based systems thinking. Based on how the rest of this section is written, it seems like Action-RESPOND is the implementation strategy. Therefore, consider moving the first mention of “(Action-RESPOND)” next to the words “multicomponent implementation strategy.”

3. I know words will be tight, but would it be possible to briefly list the 7 implementation strategies of Action-RESPOND in the methods section?

INTRODUCTION

4. Line 58: As part of the opening sentence, consider mentioning some of the health conditions associated with childhood obesity to highlight the point that obesity is a population health challenge.

5. Lines 63-64: Provide some context about how environmental and individual determinants interact in complex systems to cause obesity. Stating some specific examples would help show readers how the intervention is addressing some of the specific environmental and individual determinants.

6. Lines 76-80: Are there any specific examples that can be provided about effective partnerships and systems lending to ongoing and sustained lifestyle and environmental changes? While the contents in this paragraph sound good, it would be good to provide a tangible example of something that worked well to address obesity.

7. Lines 108-113: The authors talk about implementation science providing evidence-based tools, methods, and frameworks that that “intentionally embedding such methods into systems thinking approaches…” It would be helpful to provide some specific examples of these tools and methods so that readers can have a tangible idea about what could specifically be embedded into systems thinking approaches. Also, providing some specific examples in the introduction section would help set a better context for the aims that are listed in lines 122-124.

MATERIALS AND METHODS

8. Line 135: The authors should specify the criteria that were used to determine the rural nature of the communities (ie., how do the investigators know that an area is rural?).

9. Line 143: Please give some examples of “key partners.”

10. Line 147: Are there any results that can be reported regarding the RESPOND program? Perhaps an outcomes paper that can be cited? It would be helpful to know about the effectiveness of RESPOND to support why there is a need to build upon this program (e.g., was the program ineffective, therefore warranting Action-RESPOND?).

11. Lines 148-157: Can the authors speak to how they are going to assess if communities have naturally adopted some of the enhanced implementation support that is being provided in Action-RESPOND?

12: Lines 164-167: As I mentioned previously, please provide the criteria that were used to determine whether communities were indeed rural.

13. Line 234: Will the investigators be keeping track of the requests for local tailored support? This information will be helpful for understanding how much support is needed to incorporate implementation strategies.

14. Lines 239-241: I am a little confused as to how the implementation coach is situated within the targeted communities. Is this saying that the implementation coach lives in these communities? Since there are 10 distinct rural communities, how is it possible for one implementation coach to be situated in all of them? Consider clarifying the implementation coach’s relationship with these communities.

15. Line 379: the authors mention that follow-up will occur at approximately 9-months post-intervention. What are the implications of having a 9-month follow-up but having participants set goals that are practical and feasible to achieve in the next 9-12 months (line 321)?

16. Line 381: In this section, it would be helpful to see the actual items that are being used to assess feasibility and adoption. Further, I would recommend adding a bit more information about the scale other than the response options ranging from 1-7. For example, what does a response option of 1 indicate compared to a response of 7?

17. Lines 393-394: Please give just a little more context about the response options of the Likert scale (similar to what the authors did on line 400).

18. Lines 433-434: Please add a citation to support that it is a best practice to not conduct a formal sample-size calculation for pilot studies (Note: I agree with this, but it would be helpful for others who question this way of thinking).

19. Line 438: In the data management section: Do the authors have a plan to assess for data entry errors and the process they plan to take to rectify potential errors?

20. Line 448 (Data analysis section): I’m struggling to understand how a linear regression can be used to compare fidelity and adoption between the two groups. The control group will not be incorporating the new implementation strategies, correct? Can the authors explain this?

21. Line 463: The authors mention that the advisory group will assess suitability for progression to a fully powered trial in similar communities based on data on fidelity, adoption, and acceptability of the implementation strategies. I am curious as to which outcome the authors will use for advancing to a fully powered trial. For a fully powered trial, it seems like the authors would want to power around weight reduction/obesity prevention rather than indicators of fidelity, adoption, and acceptability of implementation strategies. In other words, it will be important to know if the authors can actually make improvements in obesity before moving forward with a fully powered trial.

22. Line 491: A citation is needed for the statement “Despite calls to better understand…”

23. As I read the discussion section, I found myself wondering how RESPOND actually addressed childhood obesity. I went back to the methods section to read about RESPOND (lines 133-147). It looks like there was routine childhood obesity and risk factor monitoring and the implementation of strategies identified in the GMB processes. It would be helpful to talk more about RESPOND in the methods section. In particular, what strategies were actually implemented to address childhood obesity?

7. PLOS authors have the option to publish the peer review history of their article (what does this mean?). If published, this will include your full peer review and any attached files.

Reviewer #1: No

Reviewer #2: No

---

## [Author Response · Author response to Decision Letter 0]

25 Feb 2024

Response to Reviewer uploaded as a document

---

## [Decision Letter · Decision Letter 1]

27 Mar 2024

A protocol of a pilot randomised trial (Action-RESPOND) to support rural and regional communities with implementing community-based systems thinking obesity prevention initiatives.

PONE-D-23-35824R1

Dear Dr. Yoong,

We’re pleased to inform you that your manuscript has been judged scientifically suitable for publication and will be formally accepted for publication once it meets all outstanding technical requirements.

Kind regards,

Sascha Köpke

Academic Editor

PLOS ONE

Reviewer's Responses to Questions

**Comments to the Author**

1. Does the manuscript provide a valid rationale for the proposed study, with clearly identified and justified research questions?

Reviewer #2: Yes

2. Is the protocol technically sound and planned in a manner that will lead to a meaningful outcome and allow testing the stated hypotheses?

Reviewer #2: Yes

3. Is the methodology feasible and described in sufficient detail to allow the work to be replicable?

Reviewer #2: Yes

4. Have the authors described where all data underlying the findings will be made available when the study is complete?

Reviewer #2: Yes

5. Is the manuscript presented in an intelligible fashion and written in standard English?

Reviewer #2: Yes

6. Review Comments to the Author

You may also provide optional suggestions and comments to authors that they might find helpful in planning their study.

Reviewer #2: I commend the authors on their thoughtful consideration of reviewer comments. I have no additional feedback. I wish the authors much success in their endeavor.

7. PLOS authors have the option to publish the peer review history of their article (what does this mean?). If published, this will include your full peer review and any attached files.

Reviewer #2: No
